# Effect of New Zealand Blackcurrant Extract on Force Steadiness of the Quadriceps Femoris Muscle during Sustained Submaximal Isometric Contraction

**DOI:** 10.3390/jfmk7020044

**Published:** 2022-05-30

**Authors:** Matthew D. Cook, Aaron Dunne, Michael Bosworth, Mark E. T. Willems

**Affiliations:** 1School of Sport and Exercise Science, University of Worcester, Worcester WR2 6AJ, UK; matthew.cook@worc.ac.uk; 2Institute of Sport, Nursing and Allied Health, University of Chichester, Chichester PO19 6PE, UK; a.dunne@chi.ac.uk (A.D.); mikebos11@hotmail.co.uk (M.B.)

**Keywords:** steadiness, blackcurrant, anthocyanins, isometric force, balance

## Abstract

Intake of anthocyanin-rich New Zealand blackcurrant (NZBC) can alter physiological responses that enhance exercise performance. In two studies, we examined the effects of NZBC extract on force steadiness during a sustained submaximal isometric contraction of the quadriceps femoris muscle. With repeated measures designs, male participants in study one (*n* = 13) and study two (*n* = 19) performed a 120 s submaximal (30%) isometric contraction of the quadriceps femoris muscle following a 7-day intake of NZBC extract and placebo (study one) and following 0 (control), 1-, 4- and 7-day intake of NZBC extract (study two). Participants for both studies were different. In study one, NZBC extract enhanced isometric force steadiness during the 120 s contraction (placebo: 6.58 ± 2.24%, NZBC extract: 6.05 ± 2.24%, *p* = 0.003), with differences in the third (60–89 s) and fourth quartile (90–120 s) of the contraction. In study two, isometric force steadiness was not changed following 1 and 4 days but was enhanced following 7-day intake of NZBC extract in comparison to control. In study two, the enhanced isometric force steadiness following 7-day intake did occur in the second (30–59 s), third (60–89 s) and fourth (90–120 s) quartiles. Daily supplementation of anthocyanin-rich NZBC extract can enhance force steadiness of the quadriceps femoris muscle during a sustained submaximal isometric contraction. Our observations may have implications for human tasks that require postural stability.

## 1. Introduction

Intake of anthocyanin-rich NZBC extract for 7 days has been shown to improve exercise performance (for reviews, see [1,2]). The possible mechanisms include increases in blood vessel diameter [3,4], blood flow [5], activity of endothelial nitric oxide synthase [6], fatigue resistance in type I muscle fibres [7], and a positive effect on the affective responses during exercise [8]. Some of these mechanisms are probably linked with the antioxidant properties of the anthocyanins [9] and the anthocyanin-derived metabolites.

Improved exercise performance following NZBC extract intake was observed for a 16.1 km cycling time trial [10], high-intensity treadmill running [11] and sports climbing [12]. For whole-body dynamic and static exercise (i.e., isometric contractions), an intervention-induced increase in exercise or contraction time demonstrates postponement of exercise-induced fatigue. However, during sustained submaximal isometric contractions, the contraction does not need to be completed to exhaustion to observe an indication of contraction-induced fatigue. During sustained submaximal isometric contractions with the task requirement for individuals to provide a constant isometric force, the isometric force shows an increase in fluctuations. The increase in isometric force fluctuations is taken as an indication of isometric contraction-induced fatigue [13,14,15].

Motor unit recruitment and activity in response to isometric contraction-induced fatigue will influence the development and profile of the force fluctuations [16]. The coefficient of variation of the force during a sustained submaximal isometric contraction provides a measure of isometric force steadiness [17]. Age [18], menstrual cycle [19], sex [20], joint position [21], neurological health [22] and resistance training [23] can affect isometric force steadiness. In general, an ability to maintain force steadiness may have applications for human tasks that require stability and control, e.g., postural stability in rhythmic gymnastics [24] and rifle shooting [25]. No studies have examined the effects of a nutritional intervention or supplement on isometric force steadiness.

In recent studies, the effects of intake duration of NZBC extract were examined on the metabolic responses during moderate-intensity walking [26,27] and femoral artery diameter during submaximal isometric contraction of the quadriceps [4]. Because anthocyanin-derived metabolites are still present in the blood 48 h following intake [28], it is possible that prolonged intake causes accumulation of metabolites and is associated with these intake-duration-dependent responses. In most studies on the effects of NZBC extract, the dosing strategy used 7 days of daily intake with 105 or 210 mg of blackcurrant anthocyanins. In Cook et al. [3], the effects of 7-day intake of NZBC extract and placebo were examined during a 2 min sustained submaximal (30%MVC) isometric contraction of the quadriceps muscles primarily for cardiovascular responses and femoral artery diameter, and the effects on isometric force steadiness were not addressed. In addition, in Cook et al. [4], intake duration (i.e., 1, 4 and 7 days) effects of NZBC extract were examined primarily on the changes in femoral artery diameter during a 2 min sustained submaximal (30%MVC) isometric contraction of the quadriceps muscles with the effects of isometric force steadiness also not addressed. These studies used an isometric force of 30%MVC to not impair blood flow due to a rise in intramuscular pressure [29].

In this manuscript, we report on the observations on isometric force steadiness from the experiments of two studies conducted in the same laboratory [3,4]. The observations from these studies support the premise that multiple days of intake of anthocyanin-rich NZBC extract are required to improve force steadiness during sustained submaximal isometric contractions.

## 2. Materials and Methods

### 2.1. Participants

For study one and two, thirteen (age: 25 ± 4 years, height: 182 ± 6 cm, body mass: 82 ± 9 kg) and nineteen physically active males (age: 26 ± 4 years, height: 178 ± 6 cm, body mass: 80 ± 6 kg) were recruited. The participants were health screened and not current smokers or habitual users of antioxidant supplements (including vitamins C and E and polyphenol-containing products). Studies were approved by the University of Chichester Research Ethics Committee with protocols and procedures performed in accordance with the ethical principles outlined by the Declaration of Helsinki (World Medical Association, 2013) with participants providing written informed consent. Ethical approval codes for studies one and two were 1617_38 and 1617_30.

### 2.2. Study One—Experimental Design and Supplementation Strategy

Study one examined the effect on force steadiness during a sustained submaximal isometric contraction by 7-day intake of NZBC extract. Study one had a placebo-controlled randomized cross-over design. Participants visited the laboratory for 3 visits at the same time of day (~8:00 am). During the first visit, height (Harpenden Wall Mounted Stadiometer, UK) and body mass (Kern ITB, Kern, Germany) were measured and participants were familiarized for all study procedures. A custom-built metal frame testing bench was used that allowed us to seat participants with hip and knee angles at ~90° with the use of goniometry. The seat settings were then replicated for remaining visits. The ankle was attached to an s-beam load cell (RS 250 kg, Teda Hutleigh Cardiff, UK) using a soft strap with a metal cuff and steel chain. Participants were firmly attached with straps across the shoulders and hips to the back support of the bench. During the voluntary isometric contractions, participants were instructed to fold arms across the chest and focus on performing a knee extension movement that could not occur due to the ankle connection with the load cell. Participants completed a warmup with three submaximal contractions (~50%MVC held for 5 s) with ~1–2 min rest between contractions. Participants then completed three maximal voluntary isometric contractions of the quadriceps muscles with two minutes of rest between contractions. Maximal voluntary isometric contractions for 3–4 s were performed with standardised instructions [30] with force measured on a computer at 1000 Hz using Chart 4 V4 1.2 (AD Instruments, Oxford, UK). The highest force for 0.5 s from the maximal voluntary isometric contractions was determined and taken as the MVC force. Subsequently, a line was placed on the computer screen displaying the 30%MVC force which participants produced for 120 s (see below for quantification of isometric force steadiness). The 120 s submaximal isometric contraction was performed ~5 min after completion of the third MVC. For all testing sessions, hip and knee angles were kept similar because force steadiness is affected by knee position [31]. For 7 days prior to experimental visits 2 and 3 for MVC and 30%MVC testing, participants consumed 2 × 300 mg capsules (total 210 mg of anthocyanins) of NZBC extract (CurraNZ^TM^, Health Currancy Ltd., Surrey, UK) or an identical looking placebo (2 × 300 mg microcrystalline cellulose M102) every morning with breakfast. Each capsule of 300 mg contains 105 mg of anthocyanins, consisting of 35–50% delphinidin-3-rutinoside, 5–20% delphinidin-3-glucoside, 30–45% cyanidin-3-rutinoside, and 3–10% cyanidin-3-glucoside. On the final day of placebo or NZBC extract supplementation (i.e., visits 2 and 3), participants reported to the laboratory two hours post-prandial of a standard breakfast (i.e., one slice of buttered bread or toast ~840 kJ, ~30 g carbohydrate, ~6 g protein and ~7 g fat) and the capsules required for that condition. A 14-day washout was used between the NZBC extract and placebo conditions, i.e., the washout was implemented on completion of visit two. Seven participants received NZBC extract as the first condition in visit two, determined by tossing a coin.

### 2.3. Study Two—Experimental Design and Supplementation Strategy

Study two examined the effect on force steadiness during a submaximal isometric contraction after 0 (control), 1-, 4- and 7-day intake of NZBC extract. Study two was designed on completion of study one and had a repeated measures design to avoid the impact of a potential learning effect in a cross-over design when participants would be tested four times first in the placebo condition. The equipment and procedures for isometric contractions were similar to study one. For study two, participants visiting the laboratory for five visits at the same time of day (8:00–10:00 am). Visit one allowed for familiarisation of the study procedures and protocols. Visit two was used for the control (baseline) condition, with visits three, four and five following 1-, 4- and 7-day intake of NZBC extract (CurraNZ^TM^, Health Currancy Ltd., Surrey, UK). The daily dosing amount was similar to in study one. Participants consumed both NZBC capsules every morning for the 7 days of intake. On days 1, 4 and 7, participants consumed both capsules two hours before arrival at the laboratory, with the same breakfast conditions as in study one. The participants were also instructed to not consume alcohol in the 24 h before the testing sessions, be well rested and consume no caffeine on the days of testing.

### 2.4. Study One and Two—Procedures for Force Steadiness

For force steadiness measurements, participants performed a 120 s submaximal (30%) isometric contraction with visual guidance of the 30% value as a line on a computer screen. Participants were instructed to perform an isometric force just above the line on the computer screen and instructed and encouraged to perform a force as constant as possible. For study one and two, isometric force steadiness was calculated for the entire 120 s contraction and for quartile sections (i.e., 0–25% (0–29 s), 25–50% (30–59 s), 50–75% (60–89 s), and 75–100% (90–120 s)) as follows:$$Isometric\;force\;steadiness=\frac{standard\;deviation\;of\;the\;force}{mean\;of\;the\;force}\times 100\%$$

### 2.5. Statistical Analysis

Statistical analysis was conducted using SPSS 26.0 (SPSS, IBM SPSS Statistics Armonk, NY, USA, IBM Corp).

#### 2.5.1. Study One

A paired samples *t*-test was used to compare maximal voluntary contraction force, average force during the 120 s contraction, and average isometric force steadiness during the 120 s contraction between placebo and NZBC extract. Isometric force steadiness between the quartile sections was determined by a condition (placebo vs. NZBC extract) by quartiles (0–25%, 25–50%, 50–75% and 75–100%), repeated measures ANOVA and Bonferroni post hoc comparison.

#### 2.5.2. Study Two

A one-way repeated measures ANOVA was used to compare average force during the 120 s contraction and the maximal isometric force 10 s following the sustained isometric contraction between control and the different intake days (i.e., 1, 4 and 7 days of NZBC extract). Isometric force steadiness was analysed by a condition (control, 1, 4 and 7-day intake) by quartiles (0–25%, 25–50%, 50–75% and 75–100%), repeated measures ANOVA and Bonferroni post hoc comparison.

#### 2.5.3. Studies One and Two

All data were assessed for normality using a Kolmogorov–Smirnov test. Homogeneity of data was assessed using Mauchley’s Test of Sphericity and when violated, Greenhouse–Geiser adjustments were made. Data are presented as means ± SD with significance accepted at *p* < 0.05. Where significant changes were observed, Cohen’s *d* effect sizes were calculated [32] with an effect size of <0.2 reported as trivial, 0.2 ≤ d < 0.5 as small, 0.5 ≤ d ≤ 0.79 as moderate and d ≥ 0.8 as large.

## 3. Results

### 3.1. Study One

The isometric forces of the maximum voluntary contractions measured following 7-day intake of NZBC extract (654 ± 73 N) or placebo (650 ± 78 N) were not different (*p* = 0.732). There was also no difference between conditions for the average force during the 120 s sustained submaximal (i.e., 30%MVC) isometric contraction (PL: 182 ± 23, NZBC extract: 182 ± 24 N, *p* = 0.934), indicating similar isometric task performance by the participants. Enhanced isometric force steadiness was observed for the complete 120 s contraction with NZBC extract (placebo: 6.58 ± 2.24%, NZBC extract: 6.05 ± 2.24%, *p* = 0.003, *d* = 0.237). With intake of NZBC extract, 11 participants (out of 13) showed enhanced isometric force steadiness. However, the quartile sections during the 120 s submaximal isometric contraction showed no time effect (F_(3, 21.670)_ = 0.957, *p* = 0.388), but there was an effect of condition (F_(1, 12)_ = 10.665, *p* = 0.007) and an interaction effect (F_(3,36)_ = 5.191, *p* = 0.04). Post hoc comparisons indicated that there was no difference in isometric force steadiness during the first (placebo: 6.5 ± 2.2%, NZBC extract: 6.5 ± 2.0%, *p* = 0.750) and second quartiles (placebo: 6.3 ± 2.0%, NZBC extract: 6.3 ± 2.0%, *p* = 0.621). There were differences during the third (placebo: 6.7 ± 2.3%, NZBC extract: 6.3 ± 2.3%, *p* = 0.040) and fourth quartiles (placebo: 6.9 ± 2.5%, NZBC extract: 6.3 ± 2.4%, *p* = 0.001) (Figure 1a). Seven-day intake of NZBC extract provided better force steadiness in the latter half of the sustained submaximal isometric contraction, suggesting that the participants were experiencing lower isometric contraction-induced fatigue.

### 3.2. Study Two

In visit one, the isometric force by the maximal voluntary contraction of the quadriceps muscles was 666 ± 145 N. For subsequent testing of submaximal sustained isometric contractions, the 30% force was based on the maximal isometric force of the first visit. There was no difference (F_(2.117,38.099)_ = 1.909, *p* = 0.160) in the average force sustained during the 120 s submaximal isometric contraction for any of the intake durations (control: 201 ± 43 N, day 1: 205 ± 40 N, day 4: 203 ± 38 N, day 7: 203 ± 40 N), indicating as in study one the similar required isometric task performance by the participants. There was also no difference (F_(3,54)_ = 0.162, *p* = 0.922) for the isometric force by the MVC performed 10 s after the 120 s isometric contraction for any of the intake duration conditions (control: 506 ± 126 N, day 1: 540 ± 130 N, day 4: 548 ± 139 N, day 7: 541 ± 138 N).

Force steadiness during the 120 s submaximal isometric contraction demonstrated a trend for a condition effect (F_(1.835,33.030)_ = 6.161, *p* = 0.06) (control: 4.96 ± 2.34%, day 1: 4.28 ± 1.69% (*d* = −0.333), day 4: 3.95 ± 1.62% (*d* = −0.502), day 7: 3.80 ± 1.60% (*d* = −0.579)). Isometric force steadiness was enhanced following 7 days of intake in comparison to 0 days (*p* = 0.004, *d* = 0.587). Isometric force steadiness in the quartiles of the 120 s contraction indicated no effect for quartiles (F_(2.020,36.351)_ = 2.083, *p* = 0.139), but there was a difference between conditions (F_(1.805,32.486)_ = 7.504, *p* = 0.003) with no interaction effect (F_(4.427,77.891)_ = 1.195, *p* = 0.320). Within the first quartile, there were no differences in isometric force steadiness between control and any of the intake durations (*p* > 0.05). During the second (*p* = 0.029, *d =* 0.707) and third quartiles (*p* = 0.023, *d* = 0.792), 7-day intake resulted in enhanced isometric force steadiness compared to control. During the last quartile, there was enhanced isometric force steadiness following 1-day (*p* = 0.026, *d* = 0.527) and 7-day intake (*p* = 0.006, *d =* 0.789) compared to control (Figure 1b). These observations suggest that 7-day intake of NZBC extract is required to alter the potential mechanisms for enhanced isometric force steadiness.

## 4. Discussion

This manuscript reports on the findings from two studies on the effects of NZBC extract on the force steadiness during a 120 s submaximal (30%MVC) isometric contraction of the quadriceps femoris muscles. The enhanced isometric force steadiness was dependent upon duration of intake, occurring following 7 days of intake, but not at 1 or 4 days. Furthermore, the force steadiness during the sustained submaximal isometric contraction did not change within the first quartile (i.e., 0–29 s). At 7-day intake in both studies, the isometric force steadiness was enhanced in the third and fourth quartiles of the contraction (i.e., 60–89 and 90–120 s). Collectively, our observations suggest that 7-day intake of NZBC extract is required to enhance isometric force steadiness. These are the first studies to examine the effects of an anthocyanin-rich supplement on force steadiness during a sustained submaximal isometric contraction. As far as we know, the effects of nutritional interventions on force steadiness during muscle contractions are absent from the literature. Previous studies have demonstrated the effect of intake duration of NZBC extract on the femoral artery diameter and metabolic responses [4,26]. Use of a placebo-controlled, cross-over design in the present study with familiarization of testing procedures probably negated that a learning effect contributed to the enhanced isometric force steadiness. Practice effects were examined for concentric and eccentric contractions in Chung-Hoon et al. [33], but without the development of muscle fatigue. In a reproducibility study by Blomkvist et al. [34], a learning effect was present for handgrip force steadiness for 5%, 10% and 25%MVC contractions that were 20 s in duration with one week between trials. In Blomkvist et al. [34], participants were not familiarized. In our studies, the participants had one full familiarization, which may have blunted a potential confounding learning effect. However, the number of trials that are needed to avoid the potential influence of a confounding learning effect when examining the effect of a supplement on isometric force steadiness is not known. Therefore, we cannot exclude that, in study two, some learning effect was partly contributing to enhanced isometric force steadiness. Future research may want to use a parallel group design to avoid the potential confounding learning effect on isometric force steadiness when tested repeatedly over a one-week period. Because of differences in isometric force steadiness between the placebo and NZBC extract conditions in study one, study two was designed to demonstrate changes by intake duration. Furthermore, the changes on day 7 in study one were not due to acute intake effects but are the result of the intake on the preceding days. If acute responses to NZBC extract occur, then this would have been expected to occur on day 1 and 4 in study two.

The mechanisms for the enhanced isometric force steadiness in the present studies may have been caused by multiple effects. Firstly, changes in local blood flow may have influenced contractile function. For example, Cook et al. [3,4] demonstrated increased femoral artery diameter from a 7-day intake of NZBC extract during a 120 s submaximal (30%MVC) isometric contraction of the quadriceps femoris muscles. Secondly, the enhanced isometric force steadiness may relate to central nervous system (CNS) functioning from neurotransmitters or changes in gut microbiota modulation. For example, Marques et al. [35] demonstrated that blackberry anthocyanin-rich extract in Wistar rats decreased tryptophan in faecal samples. As tryptophan is a precursor to serotonin and kynurenine, its availability may influence the CNS. In addition, Henderson et al. [36] demonstrated that enhanced serotonin availability decreases physical tremor amplitude and improves steadiness during a sustained isometric contraction of the elbow flexors with the presence of fatigue. Interestingly, Lomiwes et al. [8] demonstrated that during moderate intensity walking, blackcurrant juice was able to lower the perception of exertion, with a strong trend (*p* = 0.06) for an increase in perceived mood, alongside an inhibition of the neurotransmitter platelet monoamine oxidase-B. Thirdly, synaptic input to motoneurons affects isometric force steadiness [37], so it seems that blackcurrant metabolites over time reduce the synaptic noise. Fourth, it is possible that contraction-induced fatigue may have been lower by the antioxidant effect of intake of NZBC extract [38].

Isometric force steadiness is associated with physical performances such as walking in individuals with multiple sclerosis [39], manual dexterity via grooved pegboard in young men and women [40], and sway during standing [41]. As the amplitude of force steadiness during isometric contractions is predictive of performance in dynamic tasks, it is possible that an improvement in force steadiness from NZBC extract may have beneficial effects in these tasks and this is something that future research could examine. Study two demonstrated effects on isometric force steadiness after 7-day intake of NZBC extract, but not at 4-day intake. The changes observed in previous studies following 7-day intake of NZBC extract are likely in response to a build-up of anthocyanin-derived metabolites. The metabolites of anthocyanins are present in blood longer than the parent anthocyanin molecule [42]. It is, therefore, likely that the anthocyanin-derived metabolites alter the mechanism(s) that can enhance isometric force steadiness.

### 4.1. Limitations

A limitation of the present studies is the absence of measurement of performance in dynamic tasks that would be influenced by an increased force steadiness. This would increase the practical significance and application of the findings from these studies and is something for future research to examine. Another limitation of the studies within this report was that electromyography was not measured in both studies. Using spike triggered averaging or spectral analysis could have provided insights into the mechanisms that enhanced isometric force steadiness.

### 4.2. Conclusions

A 7-day intake of NZBC extract causes an increase in force steadiness during a 120 s isometric contraction of the quadriceps muscles at 30% of their maximal voluntary contraction force. The increased isometric force steadiness occurs after 7-day intake of NZBC extract, but not after 1- or 4-day intake.

## Figures and Tables

**Figure 1 jfmk-07-00044-f001:**
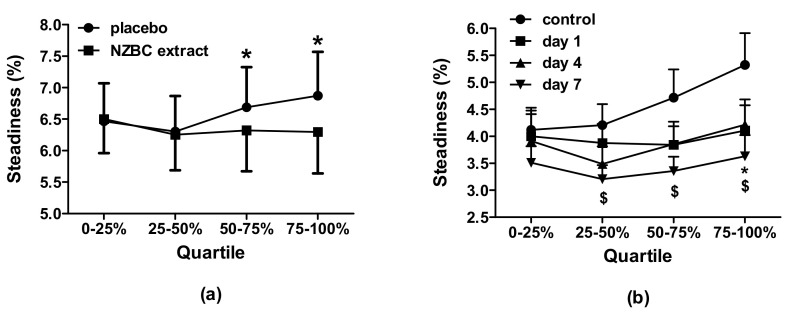
Isometric force steadiness in the quartile sections during a 120 s submaximal (30%MVC) isometric contraction of the quadriceps muscles: (**a**) Effects of 7-day intake for placebo and NZBC extract, * = difference between placebo and 7-day intake of NZBC extract in the 3rd and 4th quartiles; (**b**) Effects of 1-, 4- and 7-day intake of NZBC extract and control (no intake), * = difference between 1 day and control, $ = difference between 7 days and control. NZBC, New Zealand blackcurrant.

## Data Availability

Data may be made available upon reasonable request.

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
