# Peer review of "Effect of New Zealand Blackcurrant Extract on Force Steadiness of the Quadriceps Femoris Muscle during Sustained Submaximal Isometric Contraction"

_jfmk, 2022, doi:10.3390/jfmk7020044_

Round 1

Reviewer 1 Report

This manuscript covers a topic of interest to muscle physiologists who are interested in performance: increases in strength performance (in this case, force steadiness) in response to consumption of a dietary supplement. The study is well-designed. Interestingly, 7 days of supplementation with New Zealand black currant extract increased force steadiness, but it did not increase other parameters of muscle performance such as maximal voluntary isometric contraction. In addition, I recommend a careful proofread by a native English speaker with considerable publication experience, as there are multiple errors that need to be addressed for this manuscript to be publication-ready (for example, see the Minor Comments).

Major comments:

  1. Last sentence of abstract: Either the conclusion is vastly overstated or the authors do not state what is meant here. Rather than “is required to”, I suggest “can”. Mechanisms can certainly be altered without daily consumption of the anthocyanins—however, as the paper does not examine mechanisms, this statement does not make sense here. Moreover, “daily presence” does not imply consumption. Replace “presence” with “supplementation”.
  2. Section 2.3: it is unclear why there is not an equivalent control group for Study two. Please clarify.
  3. Line 207: “Seven days intake of New Zealand blackcurrant extract reduces the mechanisms responsible for contraction induced fatigue during submaximal sustained isometric exercise.” This paper only examines physiological measures of muscle contraction and does not explore the mechanisms by which anthocyanins may affect muscle contraction. Moreover, was fatigue per se measured? Alternatively, do the data suggest that fatigue was reduced?
  4. Lines 209-11: This sentence doesn’t seem to belong here. This is the paragraph discussing the results of Study one.
  5. For Figure 2, I strongly suggest using a connected line graph rather than a bar graph for each of the panels, as the data indicate changes over time.
  6. In the introduction (lines 36-38), it suggests that potential performance-enhancing benefits of New Zealand blackcurrant extract “…are probably linked with…antioxidant properties…” However, in the discussion (paragraph starting on line 278), it is unclear how the proposed mechanisms are related to antioxidant activity. Please clarify.

Minor comments:

  1. When used as adjectives, nouns should take the singular form. Thus, at all instances using “days intake”, the appropriate form should be “day intake”. For instance, “A 7-day intake of mushrooms alters mood.”
  2. Hyphens for “7 day” or “7 days” should only be used when applied as an adjective. For instance “intake for 7 days” and “7-day intake”.
  3. Some of the language of the manuscript is quite wordy. I suggest a careful revision focusing on reducing wordiness. As an example, on line 45: “be completed to exhaustion to observe an indication of the development of…” It seems that “an indication of the” could be deleted without changing the meaning of the sentence. This would avoid the use of four consecutive prepositional phrases.
  4. The abbreviation “NZBC” is not used consistently throughout the manuscript. After the abbreviation is defined at the first instance, only the abbreviation should be used thereafter. Accordingly, all instances of “New Zealand blackcurrant” that occur after the first instance should be replaced with “NZBC”.
  5. Sentence ending on line 112 lacks a period.
  6. Line 167 and 174: delete “participants”. It is not needed and reads awkwardly.
  7. Why is “post hoc” italicized on line 179 but not on line 172 (or elsewhere)? Be consistent with presentation.
  8. The full sentence on line 198 is not a complete sentence as written.
  9. Line 295: “overtime” should be “over time”.

Author Response

We thank the reviewer for the time and anticipation to review the manuscript. Your comments and suggestions have allowed us to strengthen and clarify the manuscript. We are also appreciative of pointing out the required grammatical corrections. Below are the point-by-point replies.

This manuscript covers a topic of interest to muscle physiologists who are interested in performance: increases in strength performance (in this case, force steadiness) in response to consumption of a dietary supplement. The study is well-designed. Interestingly, 7 days of supplementation with New Zealand black currant extract increased force steadiness, but it did not increase other parameters of muscle performance such as maximal voluntary isometric contraction. In addition, I recommend a careful proofread by a native English speaker with considerable publication experience, as there are multiple errors that need to be addressed for this manuscript to be publication-ready (for example, see the Minor Comments).

Reply. Thanks for the recognition of the topic of the paper. Three of the authors are native English and we have carefully proofread and made numerous changes, which were also requested by another reviewer.

Major comments:

  1. Last sentence of abstract: Either the conclusion is vastly overstated or the authors do not state what is meant here. Rather than “is required to”, I suggest “can”. Mechanisms can certainly be altered without daily consumption of the anthocyanins—however, as the paper does not examine mechanisms, this statement does not make sense here. Moreover, “daily presence” does not imply consumption. Replace “presence” with “supplementation”.

Reply. “is required to” is replaced with “can”. We have made the conclusion more specific to avoid we examined mechanisms with focus now as well on the implication of our findings. We added “Our observations may have implications for human tasks that require postural stability. We replaced “presence” with “supplementation”.

  1. Section 2.3: it is unclear why there is not an equivalent control group for Study two. Please clarify.

Reply. Study two was designed after completion of study one. We had the concern of a learning effect with a cross-over study in which the participants would take placebo first and perform a number of steadiness test, and acknowledge our approach as a limitation. In a reproducibility study by Blomkvist et al (2018), a learning effect was present for handgrip force steadiness for 5%, 10% and 25%MVC contractions that were 20 seconds in duration with one week between tests. In Blomkvist et al (2018), participants were not familiarized. The absence of familiarization may have affected the learning effect. In our studies, the participants had one full familiarization which may have blunted a potential learning effect when tested for placebo and NZBC extract with 2 weeks between testing. In study one, the use of a cross-over design should have avoided an interference of a learning effect and 7 participants received NZBC extract as first condition.

We added the following “In a reproducibility study by Blomkvist et al (2018), a learning effect was present for handgrip force steadiness for 5%, 10% and 25%MVC contractions that were 20 seconds in duration with one week between trials. In Blomkvist et al (2018), participants were not familiarized. In our studies, the participants had one full familiarization which may have blunted a potential confounding learning effect. However, the number of trials that are needed to avoid the potential influence of a confounding learning effect when examining the effect of a supplement on isometric force steadiness is not known. Therefore, future research may want to use a parallel group design to avoid the potential confounding learning effect on isometric force steadiness when tested repeatedly over a one-week period”.  

  1. Line 207: “Seven days intake of New Zealand blackcurrant extract reduces the mechanisms responsible for contraction induced fatigue during submaximal sustained isometric exercise.” This paper only examines physiological measures of muscle contraction and does not explore the mechanisms by which anthocyanins may affect muscle contraction. Moreover, was fatigue per semeasured? Alternatively, do the data suggest that fatigue was reduced?

Reply. Previous research has shown that the lower steadiness at the end of sustained contraction is indicative of contraction-induced fatigue. We agree that we do not examine the mechanism of contraction-induced fatigue but the higher steadiness in the placebo condition indicates that the participants are experiencing more contraction-induced fatigue. We have removed statements on mechanism.

We have revised to “Seven days intake of New Zealand blackcurrant extract provided better steadiness in the latter half of the sustained isometric contraction suggesting that the participants were experiencing lower isometric contraction-induced fatigue.”

“We also added in the introduction the following “The increase in isometric force fluctuations are taken as an indication of isometric-contraction induced fatigue [13-15].

Motor unit recruitment and activity in response to isometric-contraction induced fatigue will influence the development and profile of the force fluctuations [16].”

  1. Missenard, O.;Mottet, D.; Perrey, S. Factors responsible for force steadiness impairment with fatigue. Muscle Nerve 2009, 40(6), 1019-1032. doi: 10.1002/mus.21331.
  2. Contessa, P.;Adam, A.; De Luca, C.J. Motor unit control and force fluctuation during fatigue. J Appl Physiol (1985) 2009, 107(1), 235-243. doi: 10.1152/japplphysiol.00035.2009.
  3. Pethick, J.;Winter, S.L.; Burnley, M. Fatigue reduces the complexity of knee extensor torque fluctuations during maximal and submaximal intermittent isometric contractions in man. J Physiol 2015, 593(8), 2085-2096. doi: 10.1113/jphysiol.2015.284380.
  4. Hunter, S.K.;Enoka, R.M. Changes in muscle activation can prolong the endurance time of a submaximal isometric contraction in humans. J Appl Physiol (1985) 2003, 94(1), 108-118. doi: 10.1152/japplphysiol.00635.2002. 

  1. Lines 209-11: This sentence doesn’t seem to belong here. This is the paragraph discussing the results of Study one.

Reply. We deleted the sentence.

  1. For Figure 2, I strongly suggest using a connected line graph rather than a bar graph for each of the panels, as the data indicate changes over time.

Reply. We have replaced Figure 2 with connected line graph. We also decided to provide both figures next to each other for ease of comparison.

  1. In the introduction (lines 36-38), it suggests that potential performance-enhancing benefits of New Zealand blackcurrant extract “…are probably linked with…antioxidant properties…” However, in the discussion (paragraph starting on line 278), it is unclear how the proposed mechanisms are related to antioxidant activity. Please clarify.

Reply. Many thanks for pointing that out. We have added the following “Fourth, it is possible that contraction-induced fatigue may have been lower by the antioxidant effect of intake of New Zealand blackcurrant extract [Reid 2008].

The reference has been added, i.e. Reid 2008. Free radicals and muscle fatigue: Of ROS, canaries, and the IOC. Free Radic Biol Med. 2008, 44(2):169-179. doi: 10.1016/j.freeradbiomed.2007.03.002.

Minor comments:

  1. When used as adjectives, nouns should take the singular form. Thus, at all instances using “days intake”, the appropriate form should be “day intake”. For instance, “A 7-day intake of mushrooms alters mood.”

Reply. Thanks, this has been changed throughout the manuscript.

  1. Hyphens for “7 day” or “7 days” should only be used when applied as an adjective. For instance “intake for 7 days” and “7-day intake”.

Reply. Thanks, this has been changed throughout the manuscript.

  1. Some of the language of the manuscript is quite wordy. I suggest a careful revision focusing on reducing wordiness. As an example, on line 45: “be completed to exhaustion to observe an indication of the development of…” It seems that “an indication of the” could be deleted without changing the meaning of the sentence. This would avoid the use of four consecutive prepositional phrases.

Reply. Thanks, we have reduced wordiness and based on comments of another reviewer, the manuscript has been substantially revised. The manuscript should be more concise.

  1. The abbreviation “NZBC” is not used consistently throughout the manuscript. After the abbreviation is defined at the first instance, only the abbreviation should be used thereafter. Accordingly, all instances of “New Zealand blackcurrant” that occur after the first instance should be replaced with “NZBC”.

Reply. Thanks, we have made the suggested change.

  1. Sentence ending on line 112 lacks a period.

Reply. Thanks, period was added.

  1. Line 167 and 174: delete “participants”. It is not needed and reads awkwardly.

Reply. Thanks, we deleted as suggested.

  1. Why is “post hoc” italicized on line 179 but not on line 172 (or elsewhere)? Be consistent with presentation.

Reply. Thanks, we changed as suggested.

  1. The full sentence on line 198 is not a complete sentence as written.

Reply. Thanks, revised.

  1. Line 295: “overtime” should be “over time”.

Reply. Thanks, revised.

Reviewer 2 Report

General comments:

The writing is often sloppy in grammar and syntax. Revise heavily. Here is one example of many that I found by the time I wrote this comment: line 198: "With intake of NZBC extract, 11 participants showing enhanced steadiness"

This should be addressed somewhere in the introduction, but why 30% MVC specifically?

Introduction

lines 43-50: somewhere in here at least one citation is needed

around line 153: What is the importance of force steadiness?  By the end of the whole introduction, the reader has no idea why this is important or what it is relevant to.

line 66-67: is that 105-210 mg per day? or over the whole week?

Methods

line 82: What was the activity level of participants, especially in regards to resistance training?

study 1 general comment: How many days after visit 1 was visit 2? It doesn't specify if it was only 7, or more, or (hopefully not) less.

study 1 general comment: The overall timing and procedures of everything is unclear. Assuming the answer to the above question is 7 days between visit, based on what is actually written in the manuscript, then the timeline is Visit 1 on day 0 for baseline testing; randomized to group and took supplement/placebo for 7 days; visit 2 on day 7 for post test; it then says there was a 14 day washout, but never really talks about visit 3. Since it states that there were only 3 visits; and assuming the groups had to go through another week of supplementation, then it works out that on day 21 the groups would start the new condition without another test, and on day 28 have their visit 3 test. That means there's no real baseline to compare against visit 3; 21-28 days between tests is not comparable to only 7 days between tests  

line 99: How many warm up trials?

line 100: How long did the maximal muscle contractions last?

line 103: Per above comment, it is unclear how long the maximal contraction lasted, but my guess was that it was 5 seconds like the warm up trials---therefore, how are you coming up with they held 30% MVC for 2 minutes during that trial?

line 106: How do you confirm their hip and knee angles?

line 107: switched citation style

line 107: what did they do on days 2 and 3? Is this coming later in the manuscript? If so, say so.

line 117: How were these 7 chosen?

at baseline, did you control that they were 2 hours post-prandial from a standard breakfast?

line 120: small grammar thing, but I would replace the word "by" with "after"

line 127-128: daily dosing was similar or the same? If not exactly the same, then it needs to be described in full

line 136: How many warm up trials? How many is a few minutes? Especially for submaximal efforts, if a few is 2-3 minutes versus 5-7 minutes, that would make a difference.

line 137: How long did they perform the maximal contraction for?

line 140-141: How much rest between the last MVC and the 2 minutes submax trial? The sentence is grammatically awkward and took several times reading it to understand

line 149-150: How restricted?

line 159: so when you say 0-25%, is that the same as 0-30s into the contraction?

line 168: average force during the MVC trials of the 30%MVC trial?

stats procedures general comment: which trials were used as the data point for each subject? Or was an average of all trials used?

lines 179-181 and 186-188: Why using partial n2 and Cohen's d?

Results

lines 207-208: This sentence, "Seven days intake of New Zealand blackcurrant extract reduces the mechanisms responsible for contraction induced fatigue during submaximal sustained isometric exercise" is not supported. First, it has not been sufficiently shown in the intro/lit review that steadiness is a good marker for fatigue (see related comments about 2nd paragraph above). Second, this study didn't measure mechanisms, just performance.

lines 229-230: report effect size

lines 238 and 240: inconsistency of terms: in one line, you refer to 0-day, in another line, you call it control

lines 235-242 and figure 2: So to clarify, Day 7 was significantly different from day 0/control, but day 7 was not different from day 1 or day 4, in the 2nd-4th quartile?  While the cross over design in study 1 helps a little, since study 1 only does control and day 7 and not the intermediary days, especially for the control group, how do you know this isn't just a learning effect?

Discussion

line 256-257: What does this clause mean? :"as well as a requirement for the development of muscle fatigue to enhance force steadiness"

lines 257-258: This sentence is awkwardly written :"The enhanced force steadiness did happen without changes in maximal voluntary isometric contraction before or after a 120-second 30% isometric contraction"

lines 265-266: No; even the cross over design in study 1 does not rule out a learning effect. In going back into the methods, I realized how inadequately study 1 was described, and that if it was conducted as literally described, it is so flawed as to not be acceptable; therefore, I am not reading past this point.

Author Response

We thank the reviewer for the comments and suggestions that allowed us to strengthen and clarify the manuscript. We really appreciate the requested detail for clarification. Below is a point-by-point reply.

General comments:

The writing is often sloppy in grammar and syntax. Revise heavily. Here is one example of many that I found by the time I wrote this comment: line 198: "With intake of NZBC extract, 11 participants showing enhanced steadiness"

Reply. We have revised throughout the manuscript.

This should be addressed somewhere in the introduction, but why 30% MVC specifically?

Reply. We have added the following. “These studies used an isometric force of 30%MVC to not impair blood flow due to a rise in intramuscular pressure [29].”

Introduction

lines 43-50: somewhere in here at least one citation is needed

Reply. We agree and added the following four references.

Missenard O, Mottet D, Perrey S. Factors responsible for force steadiness impairment with fatigue. Muscle Nerve. 2009 Dec;40(6):1019-32. doi: 10.1002/mus.21331.

Contessa P, Adam A, De Luca CJ. Motor unit control and force fluctuation during fatigue. J Appl Physiol (1985). 2009 Jul;107(1):235-43. doi: 10.1152/japplphysiol.00035.2009.

Pethick J, Winter SL, Burnley M.Fatigue reduces the complexity of knee extensor torque fluctuations during maximal and submaximal intermittent isometric contractions in man. J Physiol. 2015 Apr 15;593(8):2085-96. doi: 10.1113/jphysiol.2015.284380.

Hunter SK, Enoka RM. Changes in muscle activation can prolong the endurance time of a submaximal isometric contraction in humans. J Appl Physiol (1985). 2003 Jan;94(1):108-18. doi: 10.1152/japplphysiol.00635.2002.

around line 153: What is the importance of force steadiness?  By the end of the whole introduction, the reader has no idea why this is important or what it is relevant to.

Reply. We added “In general, an ability to maintain force steadiness may have applications for human tasks that require stability and control, e.g. postural stability in rhythmic gymnastics [24] and rifle shooting [25].”

line 66-67: is that 105-210 mg per day? or over the whole week?

Reply. Apologies for the confusion. We have revised to “7 days of daily intake with 105 or 210 mg of blackcurrant anthocyanins”

Methods

line 82: What was the activity level of participants, especially in regards to resistance training?

Reply. The male participants were recruited from the student population primarily from students on Sport and Exercise Sciences and Sport and Exercise Physiology programmes. None was specifically involved in a structured resistance training program. We do not have a record of their physical activity levels but they are best described as physically active. This has been added.

study 1 general comment: How many days after visit 1 was visit 2? It doesn't specify if it was only 7, or more, or (hopefully not) less.

Reply. We had already the statement “A 14-day washout was used between the NZBC extract and placebo conditions.” There was therefore 21 days between visit 2 and 3. No changes were made.

study 1 general comment: The overall timing and procedures of everything is unclear. Assuming the answer to the above question is 7 days between visit, based on what is actually written in the manuscript, then the timeline is Visit 1 on day 0 for baseline testing; randomized to group and took supplement/placebo for 7 days; visit 2 on day 7 for post test; it then says there was a 14 day washout, but never really talks about visit 3. Since it states that there were only 3 visits; and assuming the groups had to go through another week of supplementation, then it works out that on day 21 the groups would start the new condition without another test, and on day 28 have their visit 3 test. That means there's no real baseline to compare against visit 3; 21-28 days between tests is not comparable to only 7 days between tests  

Reply. Thanks for this comment. A similar dosing strategy has been used before e.g. Cook et al 2015. The dosing strategy is study one, due to the length of time indeed between visit 2 and 3, i.e. 21 days. For study one, the observations in the NZBC extract condition are compared to those in the placebo condition. It seems our interpretation of 14-day washout differs. We have clarified that the washout was applied following visit 2. We added “A 14-day washout was used between the NZBC extract and placebo conditions, i.e. the washout was implemented on completion of visit two.” We hope this addresses your point.

line 99: How many warm up trials?

Reply. This has been clarified with “Participants completed a warmup with three submaximal contractions (~50%MVC held for 5 s) with a few minutes rest between contractions.”

line 100: How long did the maximal muscle contractions last?

Reply. This has been clarified with “Maximal voluntary isometric contractions for 3-4 seconds were performed…”

line 103: Per above comment, it is unclear how long the maximal contraction lasted, but my guess was that it was 5 seconds like the warm up trials---therefore, how are you coming up with they held 30% MVC for 2 minutes during that trial?

Reply. That was indeed unclear. We have clarified with “The highest force for 0.5 s from the maximal voluntary isometric contractions was determined and taken as the MVC force. Subsequently, a line was placed on the computer screen displaying the 30%MVC force which participants produced for 120 seconds….”

line 106: How do you confirm their hip and knee angles?

Reply. Participants were sitting on the testing bench which had 90 degrees between the seating part and the back support ensuring a hip angle of 90 degrees. The knee was positioned to 90 degrees with use of goniometry as back of the knee was in contact with the seating part. We have clarified this in the manuscript.

line 107: switched citation style

Reply. Thanks for spotting that. This has been changed.

line 107: what did they do on days 2 and 3? Is this coming later in the manuscript? If so, say so.

Reply. We have clarified with “For 7 days prior to the experimental visits 2 and 3 for MVC and 30%MVC testing…”

line 117: How were these 7 chosen?

Reply. We randomized the visits by tossing a coin. This has been clarified with “Seven participants received NZBC extract as the first condition in visit two, determined by tossing a coin.”

at baseline, did you control that they were 2 hours post-prandial from a standard breakfast?

Reply. No breakfast control for the first visit, but testing at the same time of day as in visits 2 and 3. We have clarified with “On the final day of placebo or NZBC extract supplementation (i.e. visits 2 and 3), participants reported to the laboratory two hours post-prandial of a standard breakfast”.

line 120: small grammar thing, but I would replace the word "by" with "after"

Reply. This has been changed.

line 127-128: daily dosing was similar or the same? If not exactly the same, then it needs to be described in full

Reply. We have clarified that the daily dose was the same as in study one.

line 136: How many warm up trials? How many is a few minutes? Especially for submaximal efforts, if a few is 2-3 minutes versus 5-7 minutes, that would make a difference.

Reply. We have clarified that it was 1-2 minutes. Thanks.

line 137: How long did they perform the maximal contraction for?

Reply. We have clarified with “The equipment and procedures for isometric contractions were similar to study one.” and we added the contraction time for the MVCs.

line 140-141: How much rest between the last MVC and the 2 minutes submax trial? The sentence is grammatically awkward and took several times reading it to understand

Reply. There was about 5 minutes between completion of the last MVC and start of the 2 minute 30%MVC contraction. This has now been clarified with “The 120-second submaximal isometric contraction was performed ~5 min after completion of the third MVC.”

line 149-150: How restricted?

Reply. We have clarified with “. During the voluntary isometric contractions, participants were instructed to fold arms across the chest and focus on performing a knee extension movement that could not occur due to the ankle connection with the load cell.”

line 159: so when you say 0-25%, is that the same as 0-30s into the contraction?

Reply. Apologies for the confusion. This has been clarified with “by quartiles [0-25% (0-29 s), 25-50% (30-59 s), 50-75% (60-89 s) and 75%-100% (90-120 s)]…”

line 168: average force during the MVC trials of the 30%MVC trial?

Reply. Thanks, this has been clarified.

stats procedures general comment: which trials were used as the data point for each subject? Or was an average of all trials used?

Reply. For each participant, the average force during the 30%MVC was measured and then the mean values calculated for the group response. This allowed us to run the statistical tests as described.

lines 179-181 and 186-188: Why using partial n2 and Cohen's d?

Reply. Partial n2 indicates effect size of the ANOVA. However, what really matters is the effect size when significance by the posthoc test. We have now only Cohen’s d.

Results

lines 207-208: This sentence, "Seven days intake of New Zealand blackcurrant extract reduces the mechanisms responsible for contraction induced fatigue during submaximal sustained isometric exercise" is not supported. First, it has not been sufficiently shown in the intro/lit review that steadiness is a good marker for fatigue (see related comments about 2nd paragraph above). Second, this study didn't measure mechanisms, just performance.

Reply. We agree. This was too strongly worded. We have clarified with “Seven-day intake of NZBC extract provided better force steadiness in the latter half of the sustained submaximal isometric contraction suggesting that the participants were experiencing lower isometric contraction-induced fatigue”. We have provided in the introduction references as requested.

lines 229-230: report effect size

Reply. Effect sizes have been added.

lines 238 and 240: inconsistency of terms: in one line, you refer to 0-day, in another line, you call it control

Reply. This has been clarified throughout the paper.

lines 235-242 and figure 2: So to clarify, Day 7 was significantly different from day 0/control, but day 7 was not different from day 1 or day 4, in the 2nd-4th quartile?  While the cross over design in study 1 helps a little, since study 1 only does control and day 7 and not the intermediary days, especially for the control group, how do you know this isn't just a learning effect?

Reply. Indeed, day 7 in study two provides differences for steadiness compared with control in the second, third and fourth quartile. We cannot be sure that there is not some learning effects. We have added as a limitation the following “In a reproducibility study by Blomkvist et al [34], a learning effect was present for handgrip force steadiness for 5%, 10% and 25%MVC contractions that were 20 seconds in duration with one week between trials. In Blomkvist et al [34], participants were not familiarized. In our studies, the participants had one full familiarization which may have blunted a potential confounding learning effect. However, the number of trials that are needed to avoid the potential influence of a confounding learning effect on isometric force steadiness is not known. Therefore, we cannot exclude that in study two some learning effect was partly contributing to enhanced isometric force steadiness. Future research may want to use a parallel group design to avoid the potential confounding learning effect on isometric force steadiness when tested repeatedly over a one-week period.”

Discussion

line 256-257: What does this clause mean? :"as well as a requirement for the development of muscle fatigue to enhance force steadiness"

Reply. Apologies, indeed not a good statement. It has been revised to “Collectively, our observations suggest that 7-day intake of NZBC extract is required to enhance isometric force steadiness.”

lines 257-258: This sentence is awkwardly written :"The enhanced force steadiness did happen without changes in maximal voluntary isometric contraction before or after a 120-second 30% isometric contraction"

Reply. We have decided to delete the statement as it may create confusion to the reader.

lines 265-266: No; even the cross over design in study 1 does not rule out a learning effect. In going back into the methods, I realized how inadequately study 1 was described, and that if it was conducted as literally described, it is so flawed as to not be acceptable; therefore, I am not reading past this point.

Reply. We agree with the reviewer that a learning effect cannot be completely ruled out. We have added “In a reproducibility study by Blomkvist et al [34], a learning effect was present for handgrip force steadiness for 5%, 10% and 25%MVC contractions that were 20 seconds in duration with one week between trials. In Blomkvist et al [34], participants were not familiarized. In our studies, the participants had one full familiarization which may have blunted a potential confounding learning effect.”

We have based on your constructive comments being able to provide a more adequate description of the procedures on study one. Similar procedures for isometric force testing was used in study two.

Thanks again for your time and anticipation.

Round 2

Reviewer 2 Report

The authors have made significant revisions in response to reviewer comments and strengthened this manuscript. Please find comments on the current version below

Introduction:

line 33: I am curious why the prior opening definition of NZBC as an acronym has been cut---if this was in response to another reviewer's comment to do so, I won't countermand them, because that's not fair, but to me that's weird.

all other changes to introduction acceptable--the purpose is much clearer

Methods

all changes to methods acceptable

Results

all changes to results acceptable

Discussion & conclusion

all changes acceptable